# Evaluation of the Expression of miR-486-3p, miR-548-3p, miR-561-5p and miR-509-5p in Tumor Biopsies of Patients with Oral Squamous Cell Carcinoma

**DOI:** 10.3390/pathogens11020211

**Published:** 2022-02-05

**Authors:** Ata Garajei, Milad Parvin, Hady Mohammadi, Abdolamir Allameh, Azin Hamidavi, Masoud Sadeghi, Azadeh Emami, Serge Brand

**Affiliations:** 1Department of Oral and Maxillofacial Surgery, School of Dentistry, Tehran University of Medical Sciences, Tehran 1416753955, Iran; atagarajei@tums.ac.ir; 2Department of Head and Neck Surgical Oncology and Reconstructive Surgery, Cancer Institute, School of Medicine, Tehran University of Medical Sciences, Tehran 1416753955, Iran; 3Department of Oral and Maxillofacial Surgery, School of Dentistry, Bushehr University of Medical Sciences, Bushehr 7514633341, Iran; m.parvin@bpums.ac.ir; 4Department of Oral and Maxillofacial Surgery, Fellowship in Maxillofacial Trauma, Health Services, Kurdistan University of Medical Sciences, Sanandaj 6617713446, Iran; hadi.mohammadi@muk.ac.ir; 5Department of Clinical Biochemistry, Faculty of Medical Sciences, Tarbiat Modares University, Tehran 1416753955, Iran; allameha@modares.ac.ir (A.A.); azihamidaviasl@yahoo.com (A.H.); 6Department of Biology, Science and Research Branch, Islamic Azad University, Tehran 1416753955, Iran; masoud.sadeghi@srbiau.ac.ir; 7Department of Anesthesiology, School of Medicine, Iran University of Medical Sciences, Tehran 1416753955, Iran; emami.a@iums.ac.ir; 8Sleep Disorders Research Center, Kermanshah University of Medical Sciences, Kermanshah 6719851115, Iran; 9Center for Affective, Stress and Sleep Disorders, University of Basel, Psychiatric Clinics, 4002 Basel, Switzerland; 10Substance Abuse Prevention Research Center, Kermanshah University of Medical Sciences, Kermanshah 6715847141, Iran; 11Department of Sport, Exercise and Health, Division of Sport Science and Psychosocial Health, University of Basel, 4052 Basel, Switzerland; 12School of Medicine, Tehran University of Medical Sciences, Tehran 1416753955, Iran

**Keywords:** oral squamous cell carcinoma (OSCC), microRNA (miRNA), real time PCR, gene expression

## Abstract

Background and objective: Oral squamous cell carcinoma (OSCC) is the most common head and neck malignancy. Expression patterns of microRNAs (miRNAs) can direct us in identifying valuable biomarkers for the prognosis of different neoplasms. Inappropriate regulation of miRNAs during physiological procedures can result in malignancies including OSCC. The aim of the present study was to evaluate the expression of miR-486-3p, miR-561-5p, miR-548-3p, and miR-509-5p in tissue biopsy samples with and without OSCC. Materials and methods: This case-control study was conducted on 17 healthy and 17 OSCC tissue biopsy samples. The expression of miRNAs was assessed using quantitative real-time PCR (q-RT-PCR) after RNA extraction from normal and cancer tissues and cDNA synthesis. Results: The means of miRNA-486-3p, miR-561-5p, and miR-548-3p expression were significantly different between OSCC and control groups (*p* < 0.001), but there was no significant difference in means of miR-509-5p expression between OSCC and control groups (*p* = 0.179). Conclusions: The findings of this study revealed that the expression of miR-486-3p and miR-561-5p was significantly lower in cancer samples compared to normal tissue samples. On the other hand, miR-548-3p expression increased in the OSCC group compared to the control group.

## 1. Introduction

Oral squamous cell carcinoma (OSCC) is the sixth most common cancer, with an annual incidence rate of more than 300,000 globally [1]. This type of cancer is the most malignant oral and maxillofacial tumor and comprises 90% of oral cancers [2,3]. The five-year survival rate of OSCC is 50%, as the majority of the patients are diagnosed at late stages [4]. Regardless of the technological and biological advancements, the prognosis of OSCC has not changed over the past decade, and the incidence of OSCC has increased. Therefore, there is a need for more advancement in the development of diagnostic and prognosis prediction tools in clinical fields.

MicroRNA (miRNA) is a small non-coding single-stranded RNA that binds to its complementary sequence, which is mainly located on three prime untranslated regions (3′-UTR) in mRNA and inhibits translocation through the RNA-induced silencing complex (RISC) [5,6]. More than 30% of human genes are regulated by miRNAs; therefore, these miRNAs control cellular, physiological, and developmental procedures and have an important role in cell proliferation, differentiation, and apoptosis [7,8]. As these procedures are deregulated in cancer, not surprisingly, many studies have been conducted to assess the relationship between miRNAs and carcinogenesis [9]. MiRNAs are generally encoded by endogenous genes and have suppressive effects on post-transcriptional regulation of their target genes by suppressing mRNA translation or degradation. This mediates the subsequent activation or blocking of downstream signaling pathways associated with oral malignancies [10]. Expression of significant miRNA differences between normal tissue, potential malignancies, and oral cancer specimens suggests that miRNA may be used as an independent prognostic marker [11,12].

Molecular markers including DNA sequencing, gene expression, and epigenetic markers have been extensively studied in the development and progression of OSCC [13,14,15,16]. However, their specificity to this type of malignancy has not been approved. Based on the more recent reports it appears that changes in the expression of small non-coding RNA such as microRNA (miRNA) during OSCC could be a more reliable marker for the cancer prognosis and diagnosis. Clinical studies clearly show a good relationship between specific miRNA expression and clinical parameters such as tumor metastasis, relapse, and survival in OSCC patients. Lin et al. [17] showed in their meta-analysis the potential of miRNAs, as OSCC diagnostic markers were assessed. The meta-analysis included eight articles; findings were mixed and inconsistent. It appeared that differences in the study design, sample sizes, sampling types, and source or population explained the mixed and inconclusive pattern of results [17]. Furthermore, findings of three meta-analyses [18,19,20] in 2018 showed some controversies regarding the miRNA expression in OSCC; the authors concluded that differences in sample characteristics might have biased the blurred and inconsistent pattern of results of their meta-analyses. As regards the geographical distribution of the studies, Troinao et al. [18] showed in their meta-analysis that 14 out of 15 studies assessed miRNAs in OSCC were performed in China, Taiwan, or Japan (East and Southeast Asia), while just one study was performed in Denmark among Europeans [21]. In addition, numerous miRNAs have been shown to be expressed differently in many studies at the OSCC, and despite the growth of the miRNA literature and their specific expression status in a variety of tissues and diseases, each new study presents a new result that ethnic differences in the subjects could be one of the main causes of these differences [22]. Therefore, there was no report from other regions, such as Iran or even any Middle-Eastern countries.

Some miRNAs have been confirmed to act as tumor suppressors in cancers such as miRNA-486-3p [23], miR-548b-3p [24], miR-561-5p [25] and miR-509-5p [26]. Based on this information, this study was designed to assess the changes in expression of specific miRNA molecules, which could be implicated in the prognosis and treatment approaches of OSCC in Iran. More specifically, changes in expression of selected miRNA, more specifically in miR-486-3p, miR-561-5p, miR-548-3p, and miR-509-5p, have been examined and compared to normal biopsies of healthy individuals.

## 2. Materials and Methods

### 2.1. Study Procedure

This case-control study was designed to assess the expression of miRNAs. To this end, seventeen tissue samples (13 males and 4 females) of patients with OSCC, and 17 tissue samples of healthy controls (13 males and 4 females) were gathered and analyzed between August 2017 and August 2018. Participants were selected from patients who were referred to the Shariati and Sina Hospitals and the Imam Khomeini Cancer Institute (Kermanshah, Iran). Patients who had either the documented OSCC diagnosis or who based on their physical examinations were suspicious for OSCC and thus underwent histopathological assessment were included as the case group. Participants in the control group were selected from patients who were referred to the dentistry department for impacted wisdom teeth extraction. Participants were fully informed about the study aims, the voluntary study participation, and the anonymous data handling. Thereafter, all participants signed the written informed consent form. Next, among participants with OSCC, biopsies were obtained from various oral, lingual, and gingival regions. Among participants in the control condition, tooth follicle samples were collected.

### 2.2. Inclusion Criteria

For individuals with OSCC, the key inclusion criterion was the documented histopathological diagnosis of OSCC in oral biopsies. For healthy individuals, the key inclusion criteria were an impacted wisdom tooth and the lack of history for malignancy and similar lesions. Next, for all participants, the signed written informed consent was mandatory, and the minimum age of participants was 18 years.

### 2.3. Exclusion Criteria

For all participants, any history of systemic, acute, or chronic inflammatory diseases was an exclusion criterion.

### 2.4. Sociodemographic and General Health Information

Participants reported on their age (years), gender (male or female at birth), smoking history (yes, no), metastasis (yes, no), neck dissection (yes, no), and the relapse (positive, negative).

### 2.5. Total RNA and miRNA Extraction

Fresh tissue samples were cut in 0.5 cm sections and immediately placed in tubes containing 0.5 mL RNAlater^®^ buffer (Ambion, Austin, TX, USA). Then, samples were transferred to the laboratory by following the guidelines for transport and maintenance of specimens. The RNAlater was extracted using the miRNeasy mini kit (Qiagen, Hilden, Germany). Total RNA, including miRNA, extraction was performed based on manufacturer guidelines. The quality and quantity of the extracted RNA were assessed using a Nanodrop Spectrophotometer™ (2000c, Thermo Fisher Scientific, Wilmington, DE, USA) at 260 mm and 270 mm wavelengths. Overall, the concentration of extracted RNAs ranged between 200 and 1000 ng/µL. The 260/280 ratio was approximately 2.13. Then, samples were frozen at −80 °C till assessment.

### 2.6. cDNA Synthesis and q-RT-PCR

The single-stranded cDNA was synthesized after RNA extraction. Then, the expression of miR-486-3p, miR-561-5p, miR-548-3p, and miR-509-5p were assessed based on the guidelines by the miRCURY LNA™ Universal RT miRNA PCR manufacturer (Exiqon, Vedbaek, Denmark). RNA samples were primarily diluted to 5 ng/µL. The thermal protocol for cDNA synthesis in 20 µL volumes was as follows; 60 min incubation at 42 °C, 5 min incubation at 95 °C, and cooling to 4 °C. Synthesized cDNA was transported to the freezer to be stored at −20 °C. Then gene expression was evaluated based on the SYBR Green method using the ABI StepOnePlus™ Real-time PCR system (Applied Biosystems, Carlsbad, CA, USA) on 10 µL of samples. The thermal protocol included primary denaturation at 95 °C for 10 min, amplification (40 cycles at 95 °C for 10 s), and incubation at 60 °C for one minute. The housekeeping U6 small nuclear RNA (U6 snRNA) was used for normalizing the expression. The desired exact mature miRNA sequences were obtained from the miRBase website. Specific primers for target genes and housekeeping genes were designed using the Allele ID software based on SYBR Green q-RT-PCR technique. Replication products of q-RT-PCR were assessed using melting curve analysis in order to assess the specificity of replication. The real-time quantitative PCR primers used were miR-486-3p forward: 5′-GGCAGCTCAGTACAGGATAAA-3′, miR-548-3p forward: 5′-ATTGGAACGATACAGAGAAGATT-3′, miR-561-5p forward: 5′-CGCGATCAAGGATCTTAAACTTTGCC-3′, and miR-509-5p forward: 5′-TTCTCCATGGTGGTGAAGACGCCA-3′. Expression of miRNAs was identified based on the threshold cycle (CT). Relative expression was calculated using 2^−ΔΔCT^ after normalization with the reference gene.

### 2.7. Statistical Analysis

The 2^−ΔΔCT^ equation was used to assess the desired miRNA expression in tumor tissue in relation to normal tissue. The level of significance was set at alpha < 0.05. All statistical computations were performed with SPSS^®^ 22.0 (IBM Corporation, Armonk, NY, USA) for Windows^®^. We used Cohen’s d (for the independent samples *t*-test) as an effect size used to indicate the standardized difference between two means. A *p*-value less than 0.01 was statistically significant.

## 3. Results

This study evaluated the expression of miR-486-3p, miR-561-5p, miR-548-3p, and miR-509-5p in tissue samples from 17 patients with OSCC and 17 healthy individuals.

Table 1 provides the descriptive demographic and clinical characteristics, separately for participants in the OSCC and in the control condition. Participants’ age was 66.17 (SD = 10.47) years for those in the OSCC condition, and 69.10 (SD = 60.46) years for those in the control condition. Gender distribution was thirteen males and four females in each group. In the OSCC group, eight (47%) were smokers; in the control condition, six (35%) were smokers. In the OSCC condition, ten (59%) had metastasis, and seven out of seventeen had a history of neck dissection.

### 3.1. Evaluation of the Expression (ΔCT) of miR-486-3p, miR-561-5p, miR-548-3p, miR-509-5p in the OSCC and Control Condition

Table 2 provides the descriptive and statistical indices of miR-486-3p, miR-561-5p, miR-548-3p, and miR-509-5p between the OSCC and control condition. Compared to the control condition, the OSCC condition showed lower miR-486-3p and miR-561-5p levels, and higher miR-548-3p levels. No differences were observed for miR-509-5p. Figure 1 provides the graphical summary.

### 3.2. Correlational Associations between the Expression of miR-486-3p, miR-561-5p, miR-548-3p, and miR-509-5p Together

Table 3 provides the correlational associations between miR-486-3p, miR-561-5p, miR-548-3p, and miR-509-5p. Higher miR-486-3p values were statistically significantly associated with higher miR-561-5p values and lower miR-548-3p values. Higher miR-561-5p values were statistically significantly associated with lower miR-548-3p values. MiR-509-5p values were statistically unrelated to miR-486-3p, miR-561-5p, and miR-548-3p values.

### 3.3. Sensitivity and Specificity

The receiver operating characteristic (ROC) curve was used to assess the sensitivity and specificity of miRNAs (Figure 2). The ROC curves of miR-486-3p, miR-561-5p, miR-548-3p, and miR-509-5p revealed the probability of them as valuable biomarkers with area under curves (AUCs) of 1.000, 0.983, 0.114, and 0.294, respectively. The ROC curve revealed that, based on AUC, miR-486-3p had the highest sensitivity and specificity, and the least sensitivity and specificity were observed for miR-548-5p.

## 4. Discussion

The key findings of the present study demonstrated that analysis of tumor tissues from Iranian patients suffering from OSCC showed that the expression of miR-486-3p and miR-561-5p were significantly lower in OSCC patients compared to controls, whereas, the expression of miR-548-3p was higher in OSCC patients compared to controls. The present results add to the current literature in an important way, because this preliminary data on changes in expression of miRNA are important in better understanding their target molecules and downstream pathways in OSCC. Further, the results have clinical and practical importance because such significant differences in certain miRNA are promising in the prognosis and treatment approaches of OSCC. With regards to clinical applications of miRNA in personalized medicine, specific miRNA expression profiles have been reported to predict specific clinical outcomes of dental implants; thus, such knowledge may be used as biomarkers in implant dentistry for diagnostic and prognostic purposes.

Nowadays, the studies evaluated new inhibitors [27] and antitumoral factors [28] in head and neck cancer. OSCC is the most common head and neck cancer and is one of the main causes of oral cancer-related mortality in the world due to its poor prognosis. Regardless of the accessibility of diagnostic and treatment modalities, the survival rate of OSCC is still very low. Therefore, providing personalized treatments can be beneficial for OSCC patients. Biomarkers with adequate precision and accuracy can be beneficial for physicians in predicting the prognosis of the disease. Therefore, physicians can choose the type and extent of treatment modalities, including surgery, chemotherapy, radiotherapy, and adjuvant therapy based on OSCC molecular profiles [18].

After selecting specific miRNA based on bioinformatic data, the present analysis revealed that the mean expression of miR-486-3p, miR-561-5p, and miR-548-3p was significantly different in the OSCC group compared to the control group. There was no statistically significant difference between groups in terms of the mean expression. Furthermore, the ROC analysis was used to assess the clinical implication of the miRNAs. The findings of the present study revealed that miR-486-3p can be a suitable choice for the diagnosis of OSCC, followed by miR-561-5p and miR-509-5p.

Chou et al. [23] reported that miR-486-3p undergoes downregulation in patients with OSCC. They also reported that discoidin domain receptor 1 (DDR1), a tyrosine kinase, is upregulated in tissues and OSCC cell lines and results in the progression of tumors. They concluded that restoration of miR-486-3p expression in OSCC can result in apoptosis in OSCC cells and reduce OSCC progression [23]. Studies on the differential expression of miR-486-3p in the plasma of OSCC patients before and after surgery showed that the expression of miR-486-3p was associated with the risk of relapse 9–12 months after surgery. Therefore, miR-486-3p was considered as a tumor suppressor miRNA in OSCC [29]. A study [30] showed that miR-486-3p can adjust the sorafenib response in hepatocellular carcinoma (HCC), targeting both fibroblast growth factor receptor 4 (FGFR4) and epidermal growth factor (EGFR), and adjusts it to be a better treatment target than FGFR4 and EGFR inhibitors.

A study on 502 tumor tissue samples of head and neck SCC (HNSCC) and 44 non-tumoral adjacent tissues revealed that miR-548f-1 was upregulated, while miR-486-1 and miR-486-2 were downregulated in HNSCC [31]. In a study by Hiramoto et al. [32], miR-509-5p was found to be associated with the worsening of survival in patients with pancreatic cancer and was considered as an independent predictive marker for mortality in pancreatic cancer. The findings of a study on patients with osteosarcoma revealed that miR-509-5p acts as a tumor suppressor miRNA by targeting TRIB2 and can be considered as an inhibitory intervention in osteosarcoma [33]. A study on the relationship between miR-509-5p and tongue SCC (TSCC) reported that the expression of this miRNA is downregulated in cell lines and tissue samples of TSCC. The authors also stated that miR-509-5p can inhibit TSCC proliferation and invasion through targeting EGFR. Therefore, the miR-509-5p/EGFR axis was considered to be capable of being a new treatment target in TSCC treatment [34].

The findings of a study on non-small cell carcinoma (NSCLC) of the lung showed that miR-561-5p is unlikely related to drug-resistant carcinoma and is considered as a potential target for antineoplastic treatment [34]. The mir-561-5p/chemokine (C-X3-C motif) ligand 1 (CX3CL1)/natural killer (NK) cell axis drives HCC metastases and shows that CX3CR1^+^ NK cells act as the strong anti-tumor therapeutic factors [35]. Mir-561-5p enhances the progression of HCC and lung metastasis in vivo, without the effect of cell proliferation and invasion in vitro [36].

The findings of a study on breast cancer showed that high expression of miR-548-3p inhibited proliferation and facilitated apoptosis in cancer cells [37]. One research [38] identified that miR-548-3p attenuated the progression of colon cancer by targeting protein for Xenopus kinesin-like protein 2 (TPX2).

## 5. Conclusions

For the first time in this study, it was reported that the expression of miR-486-3p, miR-561-5p, miR-548-3p, and miR-509-5p was affected in Iranian patients with OSCC. The overall findings of this study indicated that the expression of miR-486-3p and miR-561-5p were significantly lower in OSCC patients compared to controls, while the expression of miR-548-3p was higher in OSCC patients compared to controls. This finding indicates the mechanism(s) by which certain microRNAs are involved in the development and progression of OSCC. Furthermore, changes in the expression of these miRNAs in cancer tissue can be considered biomarkers for the early detection of OSCC. These findings indicate the potential of miRNAs in the clinical assessment of cancers and provide new data for the development of molecule targeting treatments for oral cancers. On the other hand, this finding in terms of oral cancer markers revealed that expression of certain miRNAs which are specifically affected during tumor development are implicated in the diagnosis of early-stage OSCC. The clinical relevance of the presented study is restricted to the publication of promising preliminary results calling for more definitive large-scale investigations. It has been shown that mir-548-3p has decreased significantly in breast cancer, and its overexpression has inhibited proliferation and promoted apoptosis of breast cancer cells when inhibition was performed by regulating the expression of enoyl coenzyme A hydratase short chain 1 (ECHS1), showing the potential of mir-548-3p as a therapeutic target for breast cancer [37].

## Figures and Tables

**Figure 1 pathogens-11-00211-f001:**
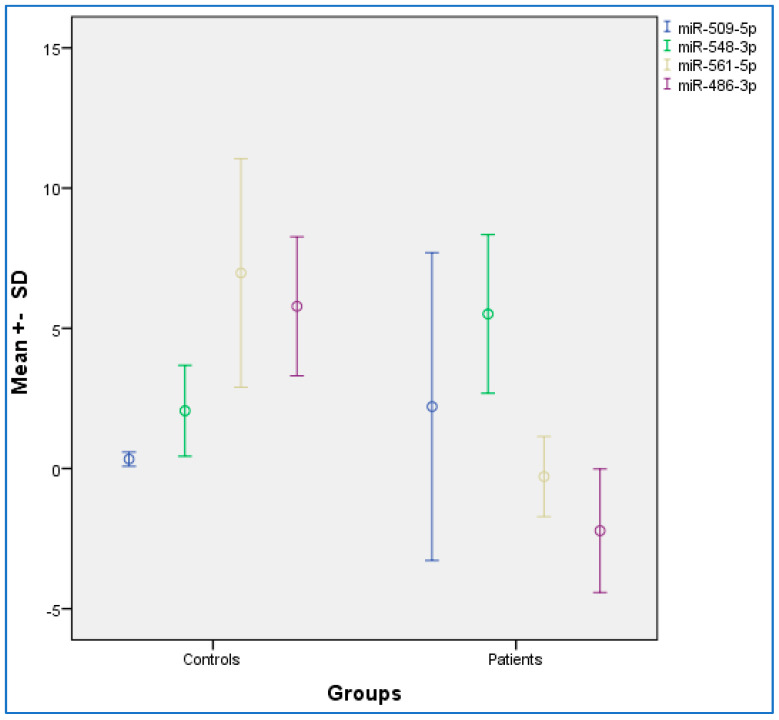
Distribution chart for the mean expression data among oral squamous cell carcinoma (OSCC) and control groups. SD, standard deviation. Error bar: 95% confidence interval.

**Figure 2 pathogens-11-00211-f002:**
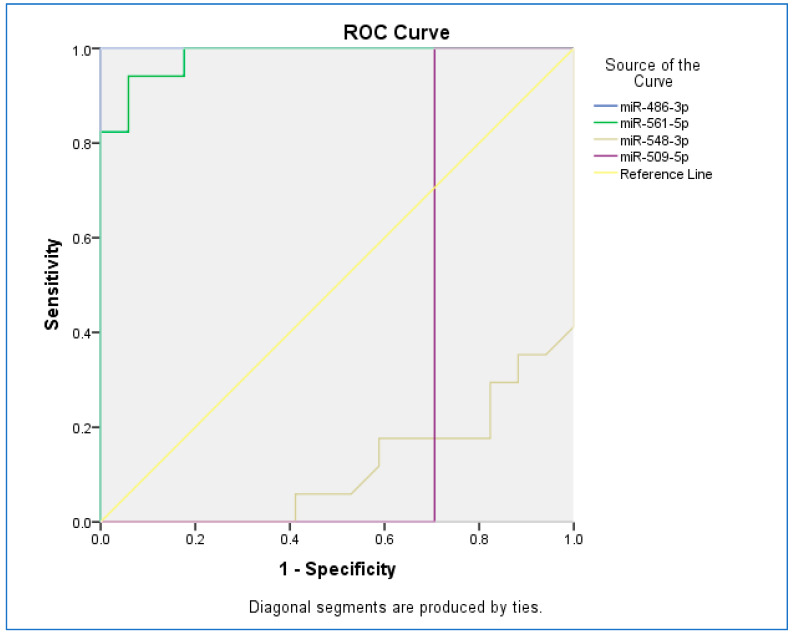
The receiver operating characteristic (ROC) curve for studied miRNAs.

**Table 1 pathogens-11-00211-t001:** Demographic and clinical characteristics of the study subjects.

Variable		OSCC Group n = 17	Control Group n = 17
Age (years ± SD)		66.17 ± 10.47	69.10 ± 60.46
Gender	Male	13 (76%)	13 (76%)
	Female	4 (24%)	4 (24%)
Relapse	Positive	7 (41%)	0 (0.0%)
	Negative	10 (59%)	17 (100.0%)
Smoking history	Yes	8 (47%)	6 (35%)
	No	9 (53%)	11 (65%)
Metastasis	Yes	10 (59%)	0 (0.0%)
	No	7 (41%)	17 (100.0%)
Neck dissection history	Yes	7 (41%)	0 (0.0%)
	No	10 (59%)	17 (100.0%)

Abbreviations: OSCC, oral squamous cell carcinoma; SD, standard deviation.

**Table 2 pathogens-11-00211-t002:** Distribution (mean ± SD) of the expression of miRNAs among OSCC and control.

Variable	OSCC Group n = 17	Control Group n = 17	*t*-Test; Cohen’s d
miR-486-3p	−2.21 ± 2.20	5.78 ± 2.47	t(32) = 9.95 ***, d = −3.65 (L)
miR-561-5p	−0.28 ± 1.42	6.97 ± 4.07	t(32) = 6.93 ***, d = −2.37 (L)
miR-548-3p	5.51 ± 2.83	2.05 ± 1.62	t(32) = −4.36 ***, d = 1.50 (L)
miR-509-5p	2.20 ± 5.49	0.33 ± 0.25	t(32) = −1.40, d = 0.48 (S)

Abbreviations: SD, standard deviation; OSCC, oral squamous cell carcinoma. Notes: *** *p* < 0.001. S = small effect size; L = large effect size.

**Table 3 pathogens-11-00211-t003:** Correlational associations between the expression of miR-486-3p, miR-561-5p, miR-548-3p, and miR-509-5p together.

		miR-486-3p	miR-561(5)	miR-548-3p	miR-509-5p
miR-486-3p	Pearson Correlation	-	0.726 **	−0.521 **	−0.187
miR-561-5p	Pearson Correlation		-	−0.553 **	−0.131
miR-548-3p	Pearson Correlation			-	0.093
miR-509-5p	Pearson Correlation				-

Notes: ** = *p* < 0.01.

## Data Availability

The datasets used and/or analyzed during the current study are available from the corresponding author on reasonable request.

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
