# Peer review of "Evaluation of the Expression of miR-486-3p, miR-548-3p, miR-561-5p and miR-509-5p in Tumor Biopsies of Patients with Oral Squamous Cell Carcinoma"

_pathogens, 2022, doi:10.3390/pathogens11020211_

Round 1
Reviewer 1 Report
The concerns were listed below;
Is there any data regarding the PD-L1 expression of the cases involved? Could this have an affect on the results of the study? Please mention the inclusion and exclusion criteria in detail for both the study and the control groups.
By writing the discussion section, please consider the hypothesis of the study. The authors have to compare their results with previous studies originating from other countries.
The conclusion section should be rewritten in order to improve the quality of the text. Please check the whole text by a native speaker.
Please avoid the use of personal pronouns such as; we, our, etc.
Please citeAçil Y, Torz K, Gülses A, Wieker H, Gerle M, Purcz N, Will OM, Eduard Meyer J, Wiltfang J. An experimental study on antitumoral effects of KI-21-3, a synthetic fragment of antimicrobial peptide LL-37, on oral squamous cell carcinoma. J Craniomaxillofac Surg. 2018 Sep;46(9):1586-1592.
and
Zhang L, Gülses A, Purcz N, Weimer J, Wiltfang J, Açil Y. A comparative assessment of the effects of integrin inhibitor cilengitide on primary culture of head and neck squamous cell carcinoma (HNSCC) and HNSCC cell lines. Clin Transl Oncol. 2019 Aug;21(8):1052-1060.
Best regards
Author Response
We thank Reviewer #1 for the care devoted to the present manuscript. The comments and suggestions helped us to improve the quality of the manuscript. Please find the detailed point-by-point-response attached as a separate file.
Again, thank you very much for all your kind efforts.

Reviewer 2 Report
Many thanks for the invitation to review the paper of Garajei and coworkers. I recommend a minor revision to the paper of Garajei et al entitled "Evaluation of the expression of miR-486-3p, miR-548-3p, miR-561-5p and miR-509-5p in tumor biopsies of patients with Oral Squamous Cell Carcinoma". The paper of Garajei et al describes the role of different miRNA (miR-486-3p, miR-548-3p, miR-561-5p and miR-509-5p) in OSCC. There are actual aspects to the oral carcinogenesis. But there are also some aspects to correct:
1)This is an original paper, not a review
2)Sample number: there are 17 OSCC and 17 oral controls (abstract), but a total of 18 in text – see line 91 (13 males and 5 females
3)Were all data collected tot he mentioned SPSS-databank – Gave the paptealso including clinical and pathological parameters?
4)Gave all patients their written consent tot he study in accordance to an approval of a local ethic committee?
5)Were all specimen fresh frozen?
6)Discussion part – which correlations to the mentioned pathways (line 204) exist? The authors focuss on DDR discoidin domain receptor – a tyrosine kinase (line 223). Some more corrleation to the role oft he investigated miRNA and DDR as well as EGFR ( line 240) would improve discussion part.
7)The clamp in line 214 should be omitted.
Author Response
We thank Reviewer #2 for the care devoted to the present manuscript. The comments and suggestions helped us to improve the quality of the manuscript. Please find the detailed point-by-point-response attached as a separate file.
Again, thank you very much for all your kind efforts.

Reviewer 3 Report
This study evaluates the expression of miR-486-3p, miR-561-5p, miR-548-3p, and miR-509-32 5p in tissue biopsy samples with and without OSCC. The study is restricted to Iranian population and has a limited number of samples, even if they are balanced in number between the two groups.
Overall the manuscript is informative, and provides new clinical and practical importance to miRNAs molecules for this malignancy, however, there are some missing experiments:
Due to the lower number of samples available, authors could check also the importance of those miRNAs in regulating some specific and known target molecules, such as mRNA molecules of genes involved in cancer progression, cell proliferation, apoptosis, etc. In this way, they can give more importance to their findings.
Authors do not explain the choice to investigate only about the expression of miR-486-3p, miR-561-5p, miR-548-3p, and miR-509-32 5p. If there is any reason, authors should explain it in the introduction.
Author should also report the primers sequence used.
- Could authors use a ROC curve to determine the incidence of metastasis in this samples basing on miRNAs expression?
In conclusion this study is interesting, but lacks of other experiments and could be improved, but I suggest a different journal for publication of this article, which is not a review.
Author Response
We thank Reviewer #3 for the care devoted to the present manuscript. The comments and suggestions helped us to improve the quality of the manuscript. Please find the detailed point-by-point-response attached as a separate file.
Again, thank you very much for all your kind efforts.

Reviewer 4 Report
General remarks
Garajei A et al have studied the expression of a set of four miRNA-s (miR-486-3p, miR-548-3p, miR-561-5p, and miR-509-5p) in 17 tissue specimens of Iranian patients with oral squamous cell carcinoma and an equal number of controls. The study is relevant as authors pointed out in the Introduction, that results of previous investigations were mixed and inconsistent (line 72). Furthermore, and hence the present study can be regarded as original, miRNA expression pattern of OSCC has not yet been determined in the Iranian population. Authors were able to demonstrate significant differential regulation of three of the four investigated miRNA-s (miR-486-3p, miR-548-3p, and miR-561-5p) in the tumor vs. the healthy control tissue.
The study and the interpretation of the results suffers, however, from a number of limitations. Authors need to explain more in detail why they chose exactly the investigated four miRNAs. In the third paragraph of Introduction, they cite four reviews and an original publication (literature items 14-18) delivering the aforementioned controversial and inconclusive results. Yet, they failed to explain how did they end up with investigating miR-486-3p, miR-548-3p, miR-561-5p, and miR-509-5p. This reviewer and colleagues have recently performed a thorough literature search in PubMed using keywords “miRNA, OSCC, and/or HNSCC” between the years of 2000 Jan and 2018 Dec. From the resulting 782 hits we selected those studies which investigated at least 20 samples using a global profiling approach and which validated their results. This way we identified a number of upregulated and downregulated miRNAs; however, miR-486-3p, miR-548-3p, miR-561-5p, and miR-509-5p was not among them (unpublished). When this reviewer repeated the literature search focusing on miR-486-3p, miR-548-3p, miR-561-5p, and miR-509-5p expression in either OSCC or HNSCC, only two hits were found, both beyond the timeframe of our original search, defining the potential role of miR-486-3p in laryngeal and oral squamous cell cc (Chou ST et al. J Exp Clin Cancer Res 2019; 38:281, Zheng Y et al. Mol Cell Biochem 2021; 476:2951). The three other miRNAs investigated in the present study resulted in 0 hits in PubMed in conjunction with keywords “OSCC” and “HNSCC”. True, using keywords “microRNA-509” and “tongue squamous cell carcinoma” (punching “squamous cell carcinoma” only was not sufficient), PubMed indicated the paper of Hou et al (literature item No 24 in the manuscript). Therefore, this reviewer does not deny that the present, quick literature "survey" might be superficial. Yet, authors need to support their choice for the selected and investigated miRNAs either by their own global miRNA profiling results or by data from the literature.
Potential mode of action of the differentially regulated miRNAs in OSCC has not been discussed in detail enough, although authors suggested in Discussion and Conclusion, that the investigated (or other? – it was not quite clear from the text) miRNAs might contribute to targeted treatment of OSCC.
Although it seems, there was no overlap in the expression levels of miR-486-3p, miR-548-3p, and miR-561-5p in the tumor vs. the healthy control tissue samples, the number of the investigated samples was rather limited to warrant the clinical relevance of the observed findings. Why should one use time consuming and expensive methods for detecting miR-486-3p, miR-548-3p, and miR-561-5p as diagnostic biomarkers, when a simple H-E staining of the biopsy sample is sufficient to decide whether or not the investigated lesion was OSCC. Possible correlations with tumor grade and stage (w/o the exception of metastatic, i.e. St IV vs. non-metastatic, i.e. St I-III lesions), as well as clinical outcome measures (with the exception of relapsing vs. non-relapsing disease) were not investigated and such efforts might have been prevented by the limited sample size. Therefore, the clinical relevance of the presented study is restricted to publication of promising preliminary results calling for more definitive large-scale investigations.
Detailed remarks
There is a typo in line 91: there should be “4” instead of “5” (13+5 equals 18 and not “seventeen”). Indeed, later in text (line 160) authors give a patient number of 13 males and 4 females.
“Study procedure” (lines 90-103): Authors should clearly state if they had obtained an institutional review board (ethical) certificate to their investigations and give the number of the certificate.
Figure 1: Authors should expand the dimension (“95% CI – or Cl? -) of the vertical (y) axis. Also, the dimension of the horizontal (x) axis should be indicated more precisely than “goupmir”. Neither abbreviations, i.e. “CI” and “groupmir” were explained either in the figure legend or in the body of the text, elsewhere.
Figure 2: A ROC graph can be called the sensitivity vs. (1 − specificity) plot, as depicted here. Drawing an imaginary line between the lower left (0,0) corner and the upper right (1,1) corner, the diagonal divides the “ROC space”. Points above the diagonal represent better than random classification results, whereas points below the diagonal represent worse than random classifiers. Therefore, the best performing biomarker, as considered miR-486-3p (marked in the light blue line in the figure) by the authors, should point to the upper left coordinate (0,1) and not to the lower right coordinate (1,0), as given in Figure 2. Also, miR-561-5p would indicate a poor performing biomarker with a low AUC value, whereas miR-548-3p and miR-509-5p were characterized by a relatively big AUC in Figure 2. Thus, results indicated by Figure 2 are just the exact opposite as written in the text of the manuscript (probably due to a faulty mathematical transformation).
Author Response
We thank Reviewer #4 for the care devoted to the present manuscript. The comments and suggestions helped us to improve the quality of the manuscript. Please find the detailed point-by-point-response attached as a separate file.
Again, thank you very much for all your kind efforts.

Round 2
Reviewer 3 Report
Authors improved the manuscript following reviewers suggestions.
The limited number of samples and patient data affect the soundness of the research but the article is informative and interesting.
Reviewer 4 Report
The applied modifications substantially improved the quality of the manuscript.
I accept all responses to the remarks made.